# Blockchain-IoT Sensor (BIoTS): A Solution to IoT-Ecosystems Security Issues

**DOI:** 10.3390/s21134388

**Published:** 2021-06-26

**Authors:** Carlos Gonzalez-Amarillo, Cristian Cardenas-Garcia , Miguel Mendoza-Moreno , Gustavo Ramirez-Gonzalez, Juan Carlos Corrales

**Affiliations:** 1Departamento de Telemática, Universidad del Cauca, Cauca 5 Nº 4-70, Popayán 190002, Colombia; jcorral@unicauca.edu.co; 2TelemaTics Research Group, Universidad Pedagógica y Tecnológica de Colombia, Tunja 150002, Colombia; cristianleonardo.cardenas@uptc.edu.co (C.C.-G.); miguel.mendoza@uptc.edu.co (M.M.-M.)

**Keywords:** IoT-Device, Blockchain-IoT ecosystem, hardware development, VHDL, food traceability

## Abstract

Sensor devices that act in the IoT architecture perception layer are characterized by low data processing and storage capacity. These reduced capabilities make the system ubiquitous and lightweight, but considerably reduce its security. The IoT-based Food Traceability Systems (FTS), aimed at ensuring food safety and quality, serve as a motivating scenario for BIoTS development and deployment; therefore, security challenges and gaps related with data integrity are analyzed from this perspective. This paper proposes the BIoTS hardware design that contains some modules built-in VHDL (SHA-256, PoW, and SD-Memory) and other peripheral electronic devices to provide capabilities to the perception layer by implementing the blockchain architecture’s security requirements in an IoT device. The proposed hardware is implemented on FPGA Altera DE0-Nano. BIoTS can participate as a miner in the blockchain network through Smart Contracts and solve security issues related to data integrity and data traceability in an Blockchain-IoT system. Blockchain algorithms implemented in IoT hardware opens a path to IoT devices’ security and ensures participation in data validation inside a food certification process.

## 1. Introduction

Internet of Things (IoT) has become an adaptable technology to any context for collecting and transmitting data, analyzing, or informing the stakeholders [1]. In all cases where IoT systems are implemented, the sensors are the basis of the system. The sensors should ensure data transparency on the communication process, from the data collection to the transmission. Nevertheless, security issues on IoT systems are mainly due to sensors’ disability for providing security on an information system.

Transparency and data integrity is a critical security issue within IoT ecosystems. Improving the communication process between peer devices might focus on solving data management from several technological schemes (fog or edge computing), but the information transparency recorded is not guaranteed. A current IoT system contains various devices with embedded sensors characterized by low power, reduced memory capacity, and limited processing capabilities. These features allow identifying the origin of significant security problems. Nevertheless, any solution concerning IoT devices’ security capabilities is designed from the software that governs the management data, generally in the system’s management layer. For this reason, new challenges arise today in the management of informatic security in communication systems.

Blockchain technology can be defined as a phenomenon from two viewpoints; financial and technological. Financial because the cooperation in a business process imposes crypto-currency and breaks the standard currency scheme [2,3,4]. Finally, technological because it also breaks the centralized systems concept and sets distributed systems to solve the critical issues of access and security [5,6,7].

Although BIoTS can act in any IoT ecosystem, we study its impact on food traceability in this paper. The traceability of Supply Chains (SC’s) is defined as the ability to track food movement through specific stages of production, transformation, and distribution [8]. Scenarios such as SC’s have found in IoT a powerful tool to trace and track products or raw materials in any production stage because it is highly interoperable, scalable, open, and ubiquitous. Moreover, its primary purpose is to keep the customer informed about the product’s harmlessness and quality with transparent data [9,10]. The main aim of traceability is to identify the origin of foods, the manufacturing process, the used ingredients, and, most significantly, to discover the responsible party or parties whenever a product is in some way faulty [11].

SC’s traceability systems enable us to locate, record, and trace products in the manufacture, processing, and distribution through platforms that offer access to users [12], for instance, IoT platforms [13]. Such media promote production quality facilitates problem identification and improves the communication capacity between stakeholders. Food security demands new concepts of trust, and blockchain is an obvious choice for further development in this regard [14].

Blockchain-IoT (BIoT) can be implemented on an SC in two ways. First, from software usage, it allows management of the network resources to share the information and reach the participation of the actors of the SC through the smart contracts [15,16]. The second way involves developing hardware immersed in an IoT sensor to enhance storage and processing capabilities. Implementing a memory module and the cryptography (SHA-256) and consensus (Proof of Work) algorithms in an FPGA is intended to provide the IoT sensor with the ability to act as a miner in the mining process in the blockchain construction [3,16,17].

This work aims to identify critical security issues concerning data integrity and transparency in IoT ecosystems’ communication process dedicated to food traceability. Besides, it proposes hardware specific to the blockchain architecture and includes it in an IoT sensor’s architecture. This device is called BIoTS (Blockchain-IoT-Sensor) with blockchain and IoT architectural features, capable of solving security issues inside an IoT network.

This article is organized as follows: Section 1 introduces the research context of this work. Section 2 presents the related works review. Section 3 details the IoT ecosystem security issues, the Blockchain-IoT on food safety context, and BIoTS-Blockchain-IoT interaction context. Section 4 presents the BIoTS architecture and a description of how a blockchain network works. Section 5 presents the BIoTS implementation details, and the cryptographic and consensus algorithms description. Section 6 presents the results and evaluation. Finally, conclusions are offered.

## 2. Related Works

According to the most frequent security vulnerabilities, in [18], they were identified and classified IoT security issues. Summarizing the security threats of IoT-Systems, some works rank these threats as challenges in the security field and propose a hierarchy of security issues; (i) low-level security issues highlight the insecure initialization, insecure physical interface, or jamming adversaries. (ii) Intermediate-level security issues highlight the insecurity of network connectivity between devices, authentication, non-secure communication on end-to-end transport-level security, and privacy violation on cloud-based IoT. (iii) High-level security issues highlight CoAP safety with the Internet, insecure interfaces, insecure Software/firmware, and middleware security. Some emerging technologies, like blockchain, can solve the majority of security problems present on IoT ecosystems, a fact that makes a new Blockchain-IoT (BIoT) concept possible.

Generally, networks based on IoT may suffer identity violation and information privacy issues, such as the services related to cloud, storage, transmission, or processing [19]. Security systems on the Internet about Constrained Application Protocol (CoAP) suffer security attacks from the application layer [20,21], this fact makes the web, mobile, and cloud interfaces vulnerable, as those indicated in (OWASP IoT top 10).

Security problems like jamming attacks are considered minor problems. Nevertheless, the message collisions and errors on the sending of the packages are solved in [22], where they measure the signal to extract the noise, then compare these measures with customized threshold measurements and detect the attack. Other solutions against a jamming attack use cryptographic functions to help correct errors [23]. Others suggest avoiding the jamming attack with encoded packages through a division of the message into blocks or changing channel frequencies for the communication flow to be successful [24].

Table 1 presents work related to solving security issues for each architecture layer of an IoT system-based traceability systems. This table presents solution alternatives with a similar approach to BIoTS for some security issues identified from the problem map in Figure 1. However, it is intended to highlight the novelty that BIoTS means in the application of blockchain in scenarios such as this one. This table describes the IoT ecosystem’s technical and technological features that BIoTS will perform to ensure product and data traceability.

The works listed in the table above define the roadmap for working in agricultural product traceability security, and helps to relate security vulnerabilities, scientific work, and threats. Technologically and conceptually, we focus on the problem and evaluate alternative solutions [40,41]. In the application layer, work is related to the deployment of a blockchain network, especially those that address security issues that impact food traceability.

Some remarkable findings for BIoTS hardware design are; a) most IoT systems’ sensors will have some features, such as; interoperability, energy, size, position, and communication. These features make the ubiquity term possible and make the system a lightweight system for they can adapt secure form between them to deploy a service in any context [42]. b) Biosensors, capable of identifying pathogens in contaminated products in the food processing industry, have gained relevance. For this reason, this proposal attempts to focus on the construction of a sensor-equipped with a technology (Blockchain) capable of guaranteeing data integrity and transparency in the transmission of information [43,44].

## 3. Foodchain Tracebility Using Blockchain

### 3.1. IoT Security Issues and Challenges

Given the range of services provided by objects, persons, or machines in the IoT networks, it is mandatory to equip both networks and devices with security features.

The standard communication protocols define the rules and security techniques in IoT networks. Fields such as health, financial security, or food safety handle processes with sensitive data that require transparency and integrity in their handling. However, the adverse factor is that as long as the design of the IoT network is done on the LLN (Low-Power and Lossy Network) network scheme, security will have that measure; that is, the device immersed in the IoT network will not have security properties beyond those possible by its capabilities.

### 3.2. Overview of Security Issues on IoT

Identifying security problems on IoT is so extensive that it generally is made from the field of application. Furthermore, the application field imposes safety criteria focused on the user and the system architecture. This hierarchy in the identification of security problems helps to identify comprehensive solutions and technologies.

Due to the technology’s capacity with which the build of BIoTS device proposed in this work, it is possible to identify the problems attending the two perspectives (Application and architecture), because of its implementation answers integrally to the IoT security problems. For this reason, this section presents the findings of some security problems, keeping a mixed approach between the security requirements of the application field and the IoT architecture.

The scheme in Figure 1 shows an IoT security issue map, which identifies in a general way the problems that affect to the IoT-based food traceability systems and focused on food safety and quality. This scheme is based on three works that propose taxonomies for identifying security issues on IoT [18,45,46,47].

Application Field Perspective: On the left of the scheme, food safety as an application field, establishes the three layers of IoT architecture as a channel to guarantee data and product traceability in the communication process throughout the IoT system.Architecture Perspective: In the upper part, the IoT architecture (Perception, Transport, and Application) establish as the central axis in the identification and classification of security problems.

The implementation of BIoTS aims to solve the security problems described here. Each layer in the IoT architecture has issues that the technology on which BIoTS solves.

Perception Layer: In this layer, where the devices that interact with the medium are hosted, we list and describe some security issues that BIoTS can solve in two scenarios; (a) in sensor nodes and (b) in sensor gateways. (a) In the sensor nodes case, to make the process of sensing and interconnecting with other nodes possible, they have these components; a controller, a transmitter (for communication), a memory where the device stores the program (code), a power source, and the hardware that obtains the sensed data. At this level, you are prone to security problems such as; node subversion, node failure, node outage, passive information gathering, false node message corruption, exhaustion, unfairness, Sybil, jamming, tampering, and collisions. (b) For sensor gateways, the collection of information on WSN represents a problem because the wireless communication channel involving radio communication and its possible appears problems such as; misconfiguration, hacking, signal loss, DoS, war dialing, protocol tunneling, man-in-the-middle attack, interruption interception, and modification fabrication. As you can see, in both cases, all security problems are aimed at attacking the trust, privacy, and integrity of the transmitted data.Transport Layer: To solve some security vulnerabilities associated with the network type BIoTS acting in a blockchain-based IoT network. The networks generally used for food traceability systems are two; (a) WiFi-centric network and (b) ad hoc non-centric network. For this reason, BIoTs and your ecosystem pretend to solve some security issues as: (a) In a WiFi network, attacks such as access attacks, malicious phishing AP, and DDos/Dos attacks. (b) In IoT, an unfocused ad hoc network is a Peer-to-Peer network. The traditional problems in this nature’s networks have to do with the communication channel’s vulnerability—attacks such as Eavesdropping, interference, vulnerable posing, cheating, Man-in-the-Middle (MitM).Application Layer: In the food safety scenario, millions of users are expected to access sensitive information on edible products. Data confidentiality and traceability is the anticipated contribution of BIoTS in the network deployed for its operation. The ecosystem is expected to contribute to security issues associated with authentication and access authorization. Besides, process safety management within a supply chain based on certification through blockchain Smart Contracts is expected.

The BIoTS sensor’s design requires scanning security issues that it can solve in the IoT context. However, the sensor can be used in a variety of fields. Next, the technological ecosystem in which BIoTS deploys its function is analyzed, i.e., the adaptability with an IoT environment, where there are also security gaps to overcome, is determined. The security problems map designed from the two points of view described in Figure 1, the design of BIoTS is focused on the issues that affect the perception layer of the IoT architecture depicted in Figure 2. Once the solutions proposed by each layer of the architecture in Table 1 are identified, the BIoTS architectural characteristics are defined according to IoT and blockchain technologies’ security requirements. Finally, the algorithms that will be designed in VHDL language for hardware construction are described.

### 3.3. Blockchain-IoT on Food Safety Context

IoT can trace or track food products, and store and process critical data for recording the process; nonetheless, it is not entirely safe in terms of data transparency held. Blockchain, beyond the crypto-currencies and the business, can certify processes transparently through data traceability.

The ability to certify the processes in food supply chains through the data integrity collected from end-to-end connection points and contained in the Blockchain-IoT architecture reveals BIoTS device’s aim. End-to-end connection points in the IoT architecture indicate the sensor’s path to the cloud service (Vertical). Simultaneously, the path from the first stage in the traceability system to the last location (Horizontal). See Figure 2.

Combining two disruptive technologies involves identifying the problems they face in applying them in food security and then finding the architectural features that make the coupling possible.

### 3.4. The Blockchain-IoT-Based Food Traceability Systems

Figure 2 shows the architectonic scheme of the Blockchain-IoT-based traceability system. The vertical levels represent three layers of IoT architecture (Perception, Transport, and Application). The six horizontal stages show the flow of agricultural products throughout the supply chain, from field to consumer. The dotted red on the Vertical levels represents the flow of data in any case. The dotted red in the horizontal stages describes the orientation of the traceability; (i) tracking the product throughout forwarding traceability, and (ii) tracing the product, aiming to know the product origin.

Blockchain technology is defined as a disruptive technology that imposes a new paradigm that can be connecting securely to the world throughout the network. Blockchain technology can be described as a platform where the transactions and the information recorded are safeguarding through cryptography algorithms in a distributed ledger to all participants of the network [48,49,50].

IoT represents an opportunity to apply blockchain technology as a support to guarantee security in some respects [51,52]. As we can see, blockchain technology is called to resolve significant problems of connecting, support, security, business, and stakeholders participating in food traceability systems (Supply Chains or Value Chains).

### 3.5. Interaction between BIoTS and Blockchain-IoT-Based Traceability System Architecture

Figure 3 shows how the BIoTS device and blockchain network interact to certify processes that depend on the information collected by BIoTS in the supply chain stages. Once the smart contract is programmed with the requirements to certify a process, in this case, humidity and temperature data, the BIoTS devices act as miners to propose a transaction and validate it within the blockchain network. In this way, the collected data will enjoy the security privileges of a blockchain system.

## 4. BIoTS Architecture

A sensor is an embedded device capable of acquiring information, processing it, analyzing it, storing it, and transmitting it to a repository. It also can coordinate with other networked devices. Under this concept, we describe BIoTS-Sensor architecture features that allow us to define the necessities of functioning to design a Blockchain-IoT system. Then, we make a description of some architectural modules.

Most sensors immerse in the IoT ecosystem, further measure some variables, and have some capacities to provide security. Nonetheless, it does not guarantee specific security requirements for low processing, energy, and storage capabilities.

These reduced capacities make the lightweight of the IoT systems and guarantee the ubiquitous characteristic of the system. However, reducing the size of the devices (sensors) present in the IoT ecosystems conflicts with the entire system’s security capabilities. For this reason, the leading security solutions presented by scientific research propose solutions on the transport layer, or (fog), and on the application layer, or (cloud), to manage security. However, the possibility of adapting the sensor hardware to an IoT architecture based on blockchain to provide the security system has not been studied so far.

Exploring this possibility has technical implications at the level of architecture and resources, for example; the system is no longer light, but there are fields of application (food and health) where the robustness of the system is worth the cost, especially if the integrity and transparency of the information are guaranteed. The BIoTS-device architecture description is made from the functional features of two technologies; IoT and blockchain. In the assembly and synchronization of both architectural blocks, it is necessary to identify the security requirements, both for software and hardware [53,54].

Block A in Figure 4 is the architectural approach of an Ethereum’s blockchain. This block describes the structural layers that form a security system. For this proposal, we focus on the Miners layer’s study and analysis. In this layer, we found the physical devices (Computers) that interact in the Network to make the blockchain valid. The fundamental elements in this layer are two; storage and processing capacity.

As we can see in Figure 4, block B represents IoT as a communication system where a service is deployed through some architectural layers Figure 2. In this block B, We focus on designing and improving the hardware capabilities in the perception layer to splice this device with block B’s blockchain software technology. The primary device features that act in this layer are; (i) interoperability, (ii) processing, (iii) energy, (iv) size, (v) position, (vi) storage capacity, and (vii) security. Moreover, it can evaluate the hardware device quality involved in an IoT ecosystem according to these features. For this reason, they are the ones we take into account to develop the new BIoTS capabilities.

The block B, we can see the module that ought added to the sensor. Each module responds to the need created by the blockchain system. The modules are related by color; thus, the blue module of the P2P Network is designed to make possible the P2P Network in which the sensor acts as Miner. The green module of the Proof of Work (PoW) algorithm makes transaction validation possible and guarantees the block’s information’s immutability. The yellow module subject to the Mining process is designed to calculate hashes in the communication’s cryptographic function. Finally, the orange module is designed to store the records for each validation through the Merkle tree.

As we can see, the junction of these blocks, the IoT as an oriented communication system, and the blockchain as a security system, together represent a communication system with a high-security level. The agro-food traceability application field, where this solution is considered, means the traceability of both product and information.

Figure 5 shows the conventional path1 used in the information flow in an IoT system and establishes the data flow and the interaction between devices within the IoT ecosystem. Path2 (Called BIoTS-Paths) sets a direct action in the information transmission, avoiding any intermediary. The BIoTS block (B) aims to make IoT hardware a miner actor in the blockchain architecture. As we can see, the objective from the hardware point of view is to make the coupling of the physical layer of the two architectures possible.

The architecture adaptation deserves to describe the hardware blocks that make the coupling possible. The following describes the Blockchain Ethereum that will govern the system and the development of the sensor module that allows it to operate in this network.

### Blockchain

BIoTS has as a challenge to adapt all its architectural modules to the functional requirements of blockchain. In this case, it is necessary to adapt two algorithms at the hardware level; (i) the SHA-256 algorithm responsible for cryptography in the communications process and (ii) the Proof of Work (PoW) algorithm responsible for the consensus process in the network.

As its name indicates, blockchain is a chain of blocks that systematically stores information in a decentralized network. Each node acts as a miner, and these, generate validation through their processing capabilities. The information contained in each block is interconnected with the previous block employing a hash, making it impossible to reverse or modify data in each block. That’s where blockchain’s security comes from Figure 6.

Figure 6 shows the principal modules contained in each block of the blockchain. The block header module contains the hash (identification on the blockchain system) for making possible interchange transactions on the network. Hash together with the nonce module; they make up a firm, part of a public and private key to transactions on the network. The block version module contains the block number (series of consecutive numbers) throughout the chain of blocks; this module serves as an identifier to know their position in the chain. The Time-Stamp module guarantees the distributed temporal database contained on each miner in the network. This module assists part of security in the system because the proof of work algorithm reads and processes it to reach the consensus. Merkle Root Hash module allows us to know the origin and history of hash blocks; this feature makes it impossible to decipher the hashes’ chain for obtaining the address or the contend any block. The nonce module assigns a zeros-chain before of hash in the block header module; thanks to this feature, each block into the blockchain has unique identification to transactions (Represents other security behavior of blockchain). Finally, the Previous Header Block module is responsible for saving the previous header hash for adding to the new hash in the new block; this module serves to form the Merkle root hash module [14].

## 5. Implementation Details

### 5.1. BIoTS Cryptographic Algorithm

A hash function can convert an input message with a specific length into an alphanumeric array on the output called a digest. A hash function has the following characteristics [56].

The reverse process of reconstructing the message from the hash is almost impossible.A minor change in the input message completely changes the output.The algorithm can compress any extension of the input message for arranging the output. It is impossible to find the same hash for two different input messages.

The SHA-256 algorithm has two modules; (i) Message Block schedule and (ii) Compression function. Below is a brief description of the modules. In the message schedule module, an N-bit message gets added with bit 1, followed by zero bits until the following, Equation (Equation 1).
(1)N+1+k=448mod512
is satisfied, where *k* indicates the number of zero bits to be added. The value *N* is then converted to its 64-bit binary representation and further added to the 448-bit intermediate value to get the 512-bit message block. This formed block is further subdivided into sixteen 32-bit word sub-blocks that input the compression function.

Compression function involves 8 registers, a,b,c,d,e,f,g,h, and 6 logical functions Ch, Maj, $\Sigma_{0}$, $\Sigma_{1}$, $\delta_{0}$, $\delta_{1}$. There are another set of eight registers, $H_{0}$, $H_{1}$, $H_{2}$, $H_{3}$, $H_{4}$, $H_{5}$, and $H_{6}$, $H_{7}$, to store 32-bit hash values, which are updated Mtimes if there are M 512-bit message blocks. These registers are initialized with 32-bit constant values obtained by considering only the fractional part of the first eight prime numbers after taking the square root. Logical functions comprise XOR, right rotation, and right shift operations. This complex operations are performed on 32-bit words for 64 rounds.

Following functions are computed on each round, and the registers are updated:Calculate Maj(a,b,c), Ch(e,f,g), $\Sigma_{0}\left(a\right)$, $\Sigma_{1}\left(e\right)$, $\delta_{0}\left(a\right)$, $\delta_{1}\left(e\right)$.Words are prepared for each round using the below equation: For the first 16 rounds,
(2)$$W_{n}=Message{{}_{n}}_{n}^{i}$$
where *n* ranges from 0 to 15 and *i* indicates number of message blocks. For the other rounds,
(3)$$W_{n}=\delta_{1}\left(W_{n-2}\right)+W_{n-7}+\delta_{1}\left(W_{n-15}\right)+W_{n-16}$$Six registers *b*, *c*, *d*, *f*, *g*, and *h* are updated with the previous registers value, i.e., *a*, *b*, *c*, *e*, *f, g*, respectively, after each round of operation. While register *a* = *T*${}_{1}$ + *T*${}_{2}$ and register *e* = *d* + *T*${}_{1}$.$T_{1}$ and $T_{2}$ have the following equations:
(4)$$T_{1}=h+\Sigma_{1}\left(e\right)+Ch+W_{n}+K_{n},$$*K* are a set of 64 constant words.
(5)$$T_{2}=h+\Sigma_{0}\left(a\right)+Maj$$After 64 rounds of operation, registers $H_{1}$ to $H_{7}$ are updated for i ranging from 1 to *M* as follows:
$$\begin{array}{c} H{}_{0}^{i}=H{}_{0}^{i-}1+a\\ H{}_{1}^{i}=H{}_{1}^{i-1}1+b\\ H{}_{2}^{i}=H{}_{2}^{i-2}1+c\\ H{}_{3}^{i}=H{}_{3}^{i-3}1+d\\ H{}_{4}^{i}=H{}_{4}^{i-4}1+e\\ H{}_{5}^{i}=H{}_{5}^{i-5}1+f\\ H{}_{6}^{i}=H{}_{6}^{i-6}1+g\\ H{}_{7}^{i}=H{}_{7}^{i-7}1+h \end{array}$$A final 256-bit hash value is obtained by concatenating 32-bit values $H{}_{0}^{M}$ to $H{}_{1}^{M}$. $Hashdigest=H{}_{0}^{M}H{}_{1}^{M}H{}_{2}^{M}H{}_{3}^{M}H{}_{4}^{M}H{}_{5}^{M}H{}_{6}^{M}H{}_{7}^{M}$.

The SHA-256 pseudo-code Algorithm 1 in the algorithm allows for a more comfortable hardware design. Generating hashes and encrypting data will enable us to understand the software structure division and translate the functions’ flow for the hardware.
**Algorithm 1 **SHA-256.1:**for** Compression Function **do**2:    Message Schedule module (Equation (Equation 1))3:    Words are prepared for each round $M_{aj}$ (Equation (Equation 2))4:    **for** First 16 rounds $W_{n}$ **do** (Equation (Equation 3))5:        Six registers b,c,d,f,g,h are updated with the previous registers6:        K are a set of 64 constant words (Equations (4) and (5))7:        After 64 rounds of operation $H_{1}$ to $H_{7}$8:        **if** $\;\mathrm{then}H{}_{7}^{i}=H{}_{7}^{i-7}$9:              Final 256-bit Hash value is obtained by concatenating 32-bit values10:        **end if**11:        Hash digest = $H{}_{0}^{M}$ to $H{}_{1}^{M}$12:    **end for**13:**end for**

### 5.2. Consensus Algorithm Analysis for BIoTS

The consensus algorithm establishes the mining agents’ computational effort to solve the mathematical puzzle that validates transactions within a blockchain network. Consensus algorithms can be categorized into two groups; proof-based consensus and vote-based consensus. In the first case, the node wishing to join the network must demonstrate higher processing and storage capabilities than the rest of the network. In the second, each node in the network is asked to propose or validate a transaction block that will be part of the validation in the rest of the network. The final decision is made only after considering the majority’s results. Thus, some algorithms were analyzed theoretically and based on [17,57,58] to select the BIoTS algorithm, some voting-based consensus algorithms; Proof of Vote (PoV), Ripple, Delegated Byzantine Fault Tolerance (DBFT), and Proof of Trust (PoT). Furthermore, two proof-based; Implicit consensus and Proof of Work (PoW).

The most common consensus algorithms in Blockchain (Proof of Stake (PoS), Delegated Proof of Stake (DPoS), and Practical Byzantine Fault Tolerance (PBFT)) limit the BIoTS ecosystem for the following reasons:PoS: It is based on the concept of the age of the coin, this age being known as its value multiplied by the period after its creation. In other words, the longer a node has a currency, the more privileges it will obtain in the network. For this reason, the BIoTS ecosystem for food traceability systems does not require concepts of this type.DPoS: It is based on the fact that each node in the network can select tokens according to their participation. These selected tokens create new blocks one by one as assigned and get a reward. Throughout the network, the n best witnesses who has participated in the transaction’s validation and has obtained the highest number of votes are entitled to the benefit. Blockchain using DPoS is more efficient and saves more energy than PoW and PoS. However, in the BIockchain-IoT ecosystem where BIoTS is deployed, it is not expected to have enough witness nodes to validate the data BIoTS collects. A BIoTS P2P network is expected to operate with consensual data sharing.PBFT: Designed to solve transmission problems and improved to avoid exponential operations. Regarding BIoTS, it is not convenient to use it as it requires a master server to execute the validation throughout all supply chain stages.

Most Blockchain networks are decentralized, with synchronous or asynchronous communication models, and are implemented in networks of nodes where mining agents are processor-based; consensus algorithms’ behavior is subject to factors such as; Blockchain type, transaction rate, scalability, adversary tolerance model, experimental setup, latency, throughput, bandwidth, communication model, communication complexity, security attacks, energy consumption, mining, consensus category, and consensus finality. Here, we analyze some of these.

Table 2 shows some characteristics of the consensus algorithms studied for implementation in the BIoTS device. As can be seen, physical experimentation on hardware to find the performance parameters does not yet exist. However, theoretically, it is possible to establish the suitability of some of them according to the Blockchain network design.

After this analysis, it is concluded that some voting-based consensus algorithms can be very relevant for BIoTS performance in a Blockchain-IoT ecosystem. However, the assembly and comparison of the consensus algorithms on FPGA are required, and this work does not have that scope. The BIoTS architecture was based on PoW because this algorithm is robust (Used in Bitcoin). The hardware built is expected to participate in an open blockchain network with significant security challenges.

### 5.3. BIoTS Consensus Algorithm (PoW)

Proof of Work consensus algorithm is a mechanism that allows users or machines to coordinate in a distributed network. This algorithm ensures that all agents in the system can agree on a single truth source, even if some agents fail. In other words, a system with PoW is tolerant of security failures.

The process of verifying the block’s transactions to be added, organizing these transactions in chronological order in the block, and announcing the newly mined block to the entire network does not take much energy and time. The energy-consuming part solves the “hard mathematical problem” to link the new block to the last block in the valid blockchain. When a miner finally finds the right solution, the node broadcasts it to the whole network simultaneously, receiving a cryptocurrency prize (the reward) provided by the PoW protocol. We show the PoW pseudo-code Algorithm 2 [14].
**Algorithm 2 **Proof of Work.r←ab               ▹ Define variable to answer1:**for** Loop from 1 to n **do**2:    var x=n;3:    **for** n major 1 $x_{0}*y_{0}+n$ **do**4:        var added = 0;5:        **for** (var i = 0; i minor Math.abs(a); i++) added += answer; **do**6:           answer = added (n–;)7:        **end for**8:    **end for**9:**end for**

Implementing the PoW in hardware determines, among other things, the difficulty in block validation (time invested in identifying the block legitimacy), and the active participation of blockchain network miners. To the hardware block of the SHA-256 algorithm, it is necessary to adapt a function capable of adding in each round of the cryptographic encoding a string of zeros (from 4 to 18 maximum, depending on the difficulty of the algorithm) that will act as an identifier of block and the transaction in the whole blockchain.

Figure 7 describes the process carried out to add the PoW to the SHA-256 algorithm in hardware. The four stages are:Generate Random Nonce: once the clock for the hash and PoW is configured and synchronized, a 32-bit register and bus are generated. The first thing is to introduce a nonce every six cycles with a feedback delay of 12 cycles.Build Block: since this is not straightforward, we use a register to track the hash through the bus. To follow the nonce, we pass the least significant 8 bits of the 32-bit register and then pair the remaining 24-bit.Difficulty: in the second of two rounds of hashing of 64 bytes each, the header of the 80-byte block is the encrypted data space. The first round gives us the average state to insert the nonce at the beginning of the record of the second 64 bytes in the correct position.Broadcast Block: subsequently, the internal hash transformation is performed to complete the register. It is still not the full SHA-256 because it involves multiple rounds. Nevertheless, this process is iterative. Here, the VHDL code in the DE10-Nano is split into phases to discriminate the SHA-256 transformations, and then unified into one block intended to do the complete hash transmission.

When the PoW algorithm checks zeros’ existence in the hash encoded in base 64, the average and the maximum number of hashes is known to calculate the order of difficulty that increases exponentially by the expression (Equation 6). The nonces included in hashes are pseudo-random, and this feature extends the capacity of the PoW.
(6)$$h\left(\rho \right)=\alpha^{\rho }$$
where, h(ρ)= the average number of hashes required to find a valid solution α = the number of characters used in the encoding ρ = the arbitrary difficulty order.
$$\begin{array}{cc} h(\rho ) & =64^{\rho }\\ h(3) & =64^{3}=262,144 \end{array}$$

Thus, typically 262,144 hashes or less are required to mine each block while testing this algorithm. However, the difficulty is arbitrarily adjusted by modifying the ρ variable. Changing ρ changes *h* exponentially and could be used to maintain a consistent network block generation rate despite exponentially increasing computational power.

## 6. Results and Evaluation

The BIoTS device comprises a platform of peripheral analog electronics connected to the digital module designed in a reconfigurable FPGA. The digital module contains several sections; cryptographic and consensus algorithms (SHA-256 and PoW) and SD storage hardware structure. The BIoTS hardware structure contains two modules (yellow and green) and one software blue module (Build of Blockchain on Python). Figure 8 shows on the right the modules designed on the DE0-Nano FPGA; these hardware modules are the ones that make the parity and interoperability of BIoTS with a blockchain network possible. This module contains, among others, the SHA-256 Algorithm, PoW Algorithm, I2C module, and SD-CARD module. These modules will be briefly described below. The next green module shows the peripherals in analog electronics for BIoTS can interact with the media, store information collected, and become a blockchain network node. The blue area contains the software development module for the blockchain network in Python, as described in the previous section.

The three blocks mentioned above occupy 47 percent of the available logic units in the DE0-Nano FPGA, barely sufficient resources for implementation [65]. In Table 3, we describe the resources available and used by the three modules.

Figure 9 shows the block diagram that summarizes the deployment of the BIoTS architecture in VHDL. As we can see, there are three blocks; I2C, SHA-256 (That contains the PoW algorithm), and the SD-CARD architecture. BIoTS need these three blocks to process two complex algorithms and store a distributed database. Here, we show the relationship between modules and how the functions and records interact. Each of these blocks contains the configuration of the logical elements that make the assigned tasks possible.

This SHA-256 algorithm design contains the Proof of Work algorithm development, necessary to calculate the nonce and make participation in the consensus process possible. The I2C master component for single master buses, written in VHDL for use in FPGAs, has component reads from and writes to user logic over a parallel interface. It was designed using Quartus II, version 18.0. Resource requirements depend on the implementation. A design incorporating this I2C master to create an SPI to I2C Bridge is available.

The SD interfaceable with FPGA is implemented from VHDL code. Here, we implement the standard size, but electrically, all sizes work the same way. Let us focus on SD card standard size, since that is conveniently popular nowadays. To this proposal, we install an SD card of 32 GB.

BIoTS analog and digital electronics’ energy consumption is calculated based on the evaluation scenario’s functional performance. The elements that consume the most energy are; the peripheral elements: the Wi-Fi module and Bluetooth. For the digital elements: SHA-256 cryptography algorithm, PoW consensus algorithm, and the SD memory block. The BIoTS power source is a three-cell LI-PO type battery at 11.2 Volts, 20–30 c discharge, and 5000 mA/h. The BIoTS analog module consumes 220 mA/h transmitting data at a 1-min interval. The power consumption of the FPGA depends on the optimization of the configured algorithms and the read and write speed of the SD memory (Sequential Read = 90 MB/s and Sequential Write = 40 MB/s).

Under these conditions, the battery life is 45 min. However, as transactions occur, the complexity algorithm kicks in and demands more processing power from the FPGA. Thus, the power consumption is dynamic, and the FPGA performance is proportional to the blockchain network’s activity.

Figure 10 shows the physical design in its prototype version once it is moved to integrated circuit hardware in a professional Beta version. This BIoTS prototype is the result of the invention described throughout this work.

### Evaluation Scenario

The BIoTS performance is evaluated according to the configuration shows in Figure 11. As we can see, the private blockchain network in which the sensor tested consists of three nodes, two computers, and the BIoTS. The computers can propose simple transactions such as submitting a humidity and temperature value, only for the BIoTS to validate them as miners in the network. However, the importance lies in the transaction proposed by the BIoTS. The computers will validate this by comparing the humidity and temperature values indicated by an internet application. Suppose the humidity and temperature value is in the right proximity range. In that case, the transaction is validated, and the functions of BIoTS as a miner in a blockchain network are satisfied.

The difficulty level setting of the consensus algorithm is low. We do not need to process a large number of network nodes for this demonstrative purpose, nor do we need to process a large data volume. However, we expect future work to demonstrate a Blockchain-IoT network configuration where a BIoTS sensors network interacts directly.

Some values related to data transmission from BIoTS to the blockchain designed for the use case are shown in Table 4. Each time, the cycle has an estimated amount of transactions per second (tps) that the device and the network can support. In this case, the data size sent humidity and temperature information that grows in the stored data in the network’s distributed database. Finally, there is a latency associated with each information transmitted and validation process for each transaction.

These data, as shown in Table 4, show the technical behavior of BIoTS in eight successive transactions. BIoTS proposed a block in the network eight times, with humidity and temperature data validated by the nodes of the network described in Figure 11. As we can see, as the size of the data sent increases, the latency in the transmission process increases. This behavior is attributed to the consensus algorithm’s performance in this type’s network.

The graph in Figure 12 shows the performance of BIoTS in transmitting a data packet concerning the time it takes to propose a transaction on the Blockchain network. We observe that the time overhead in BIoTS transactions is one second; this is when it takes for the algorithm to encode and decode the accumulated data from seven humidity and temperature readings. The behavior of the transaction rate is linear, while the difficulty of the consensus algorithm grows exponentially. Directly, the size of the data packet sent in each transaction increases. This linear behavior may change to saturation lapses when BIoTS is subjected to the work of a blockchain network where multiple BIoTS nodes participate. However, at the same time, the network will work at the execution rate of the algorithms on FPGA. The transactions per second will surely increase, and the difference in network performance compared to a processor-based network can be determined.

The data shown in remix.ethereum.org (Figure 13) is the response to the programming of a Smart Contract in Solidity language; it is designed to read the humidity and temperature data of BIoTS every so often. These data are sent from BIoTS generating a new transaction, or if it is the case, another device proposes the transaction, and BIoTS can corroborate this information. The data shown in Figure 13 shows the behavior of BIoTS acting as a miner within the Blockchain-IoT network.

For the evaluation scenario, the transmitted humidity and temperature data from BIoTS are correctly encrypted and recorded in the blockchain network. Since the computational overhead increases as transactions (Measurements) and blocks grow, it is only possible to know the energy performance of BIoTS when hundreds or thousands of reads are accumulated. However, future work is expected to implement a BIoTS network and determine its performance as a Blockchain-IoT ecosystem.

Table 5 summarizes some of the IoT security issues that BIoTS has the potential to solve. As we can see, with this device, the resistance to specific security attacks is high and some moderate, considering the raids in a Blockchain-IoT network.

To identify the security flaws where BIoTS is a potential solution, we study a causality and effect correlation between the nature of the security attack in an IoT ecosystem and the Blockchain Hardware’s architectural characteristics implemented in BIoTS [3,14,53]. The scale is weighted according to the architecture’s characteristics. For example:“Sensor Tampering”: the attack on BIoTS is unlikely because the sensor data is hosted in the SHA-256 and PoW hardware algorithms; after this process, the information is encrypted.“Sensor Feed Modification”: this attack is possible with BIoTS; however, the resistance is high because the BIoTS firmware is almost null, and almost all elements are hardware.“Sybil Attack”: this attack is unlikely in a network where BIoTS acts because it has the same blockchain network’s resilience. However, it all depends on the network configuration (complexity level of PoW, etc.).“DoS, Protocol tunneling, and man-in-the-middle”: these attacks are unlikely due to the blockchain network’s nature. The communication channel by cryptography and the algorithms carried in hardware is immune to external intervention.“Jamming and Collisions”: these attacks are possible in a BIoTS network. The resistance to the attack is moderate because it can identify the hash’s inputs and outputs to reproduce copies.

## 7. Conclusions

This paper proposes the hardware architecture of blockchain (Cryptography and Consensus Algorithms) to enrich an IoT device and provide it with remarkable capabilities to address security issues in the IoT ecosystem. This paper proposes a secure and trusted blockchain hardware architecture thanks to the reconfigurability of an FPGA. Our goal was to realize the implementation of Blockchain in hardware adapted to an IoT device to improve physical and virtual security mechanisms on devices in a network. This proof of concept works well and can positively impact the field of IoT-based food traceability systems. Finally, IoT devices immersed in ecosystems dedicated to the security and quality of supply chains are destined to be critical actors in products’ quality and safety through complete confidence in their performance in collecting, transmission, and storing information. This concept may be the future of primary IoT devices’ capabilities in food traceability systems or other application fields. The adaptation of emerging technologies in the IT security applied to various application fields opens a path to the evolution of Blockchain-IoT systems.

## Figures and Tables

**Figure 1 sensors-21-04388-f001:**
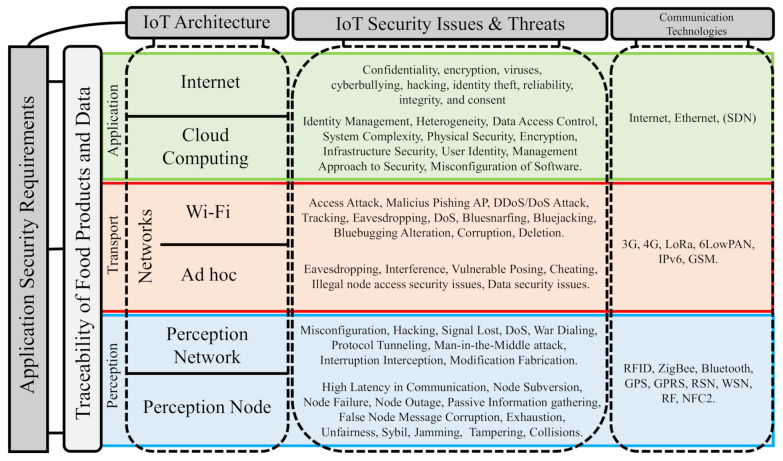
IoT security issues and threats.

**Figure 2 sensors-21-04388-f002:**
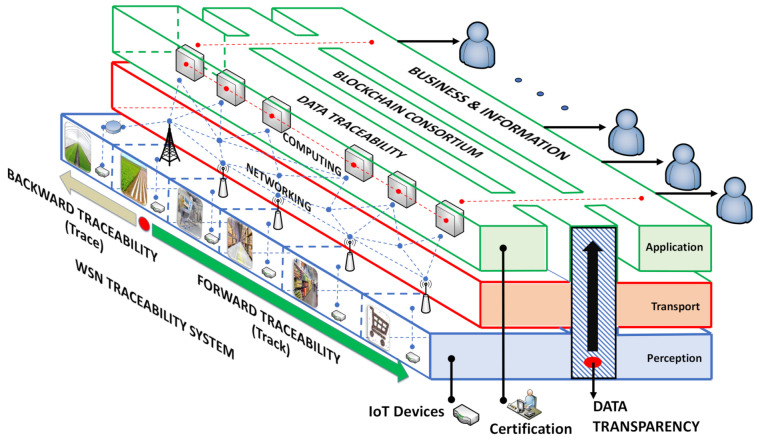
The Blockchain-IoT-based food traceability systems.

**Figure 3 sensors-21-04388-f003:**
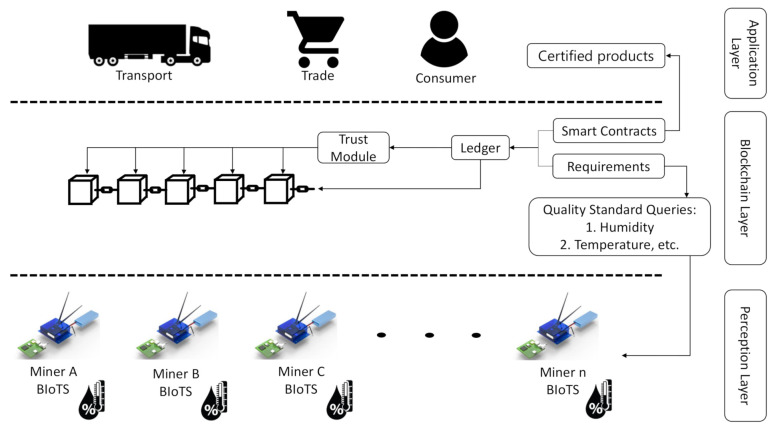
BIoTS System Operation.

**Figure 4 sensors-21-04388-f004:**
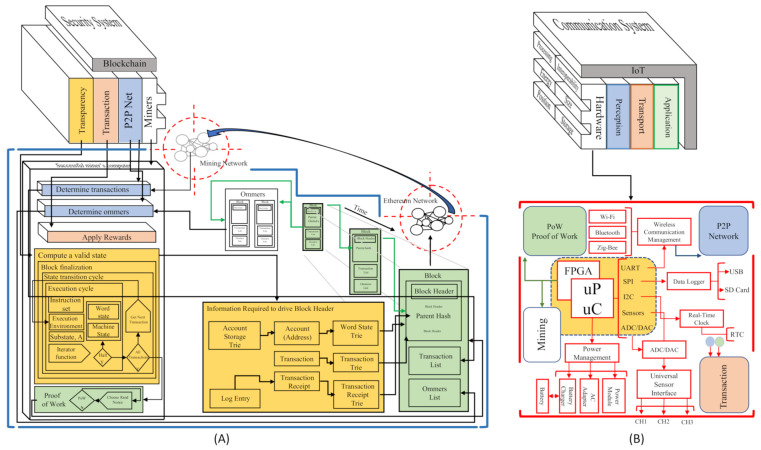
Blockchain-IoT Architecture Matching. (**A**) Blockchain Ethereum Architecture Approach by Lee Thomas based on [55]. (**B**) IoT-Sensor Architecture.

**Figure 5 sensors-21-04388-f005:**
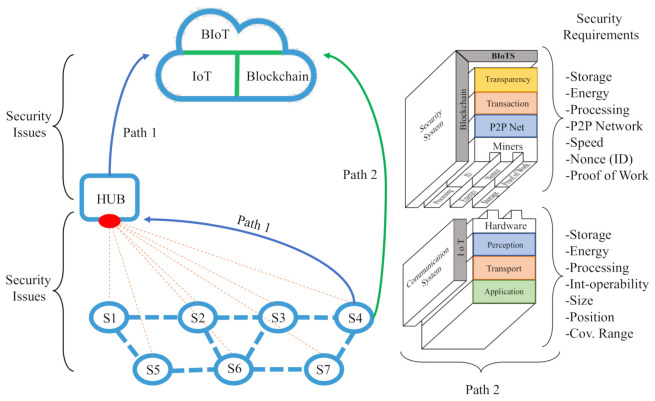
Path 1: conventional data transmission in an IoT system. Path 2: architecture and transmission path proposed by (BIoTS-Paths).

**Figure 6 sensors-21-04388-f006:**
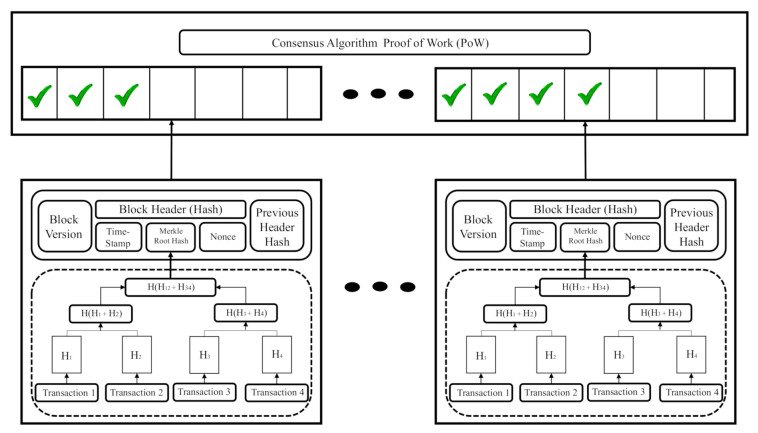
Blockchain system architecture and transaction validation mechanism.

**Figure 7 sensors-21-04388-f007:**
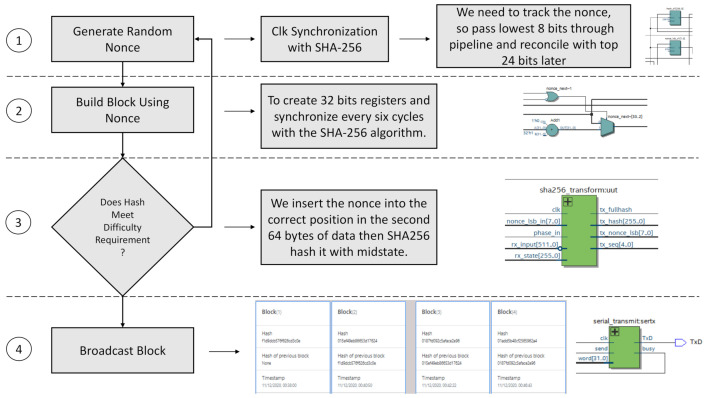
Proof of work implementation on hardware.

**Figure 8 sensors-21-04388-f008:**
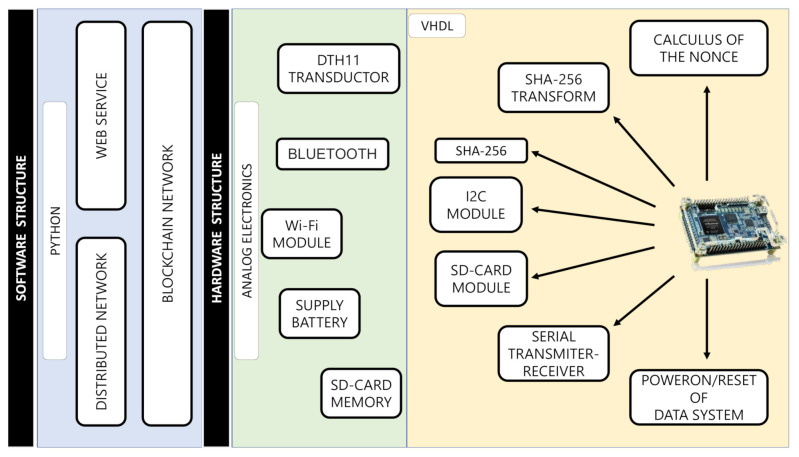
Structure of architectural development.

**Figure 9 sensors-21-04388-f009:**
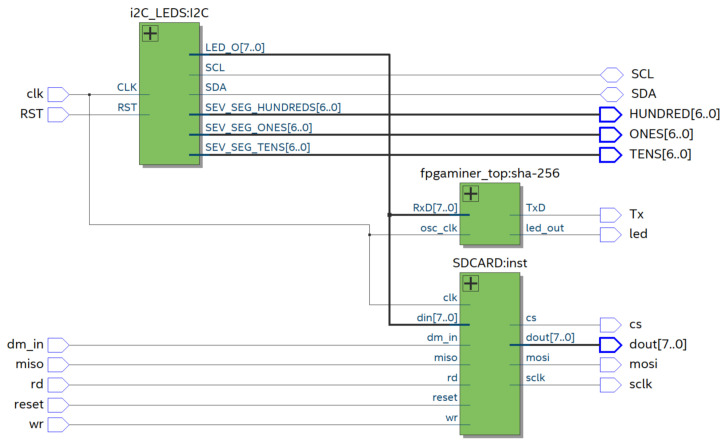
Diagram block of BIoTS.

**Figure 10 sensors-21-04388-f010:**
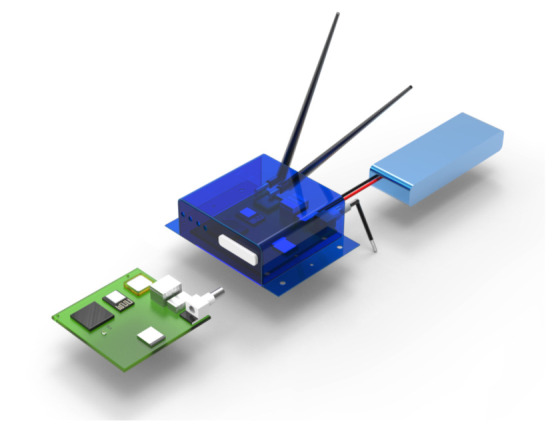
Full-scale modeling of BIoTS prototype.

**Figure 11 sensors-21-04388-f011:**
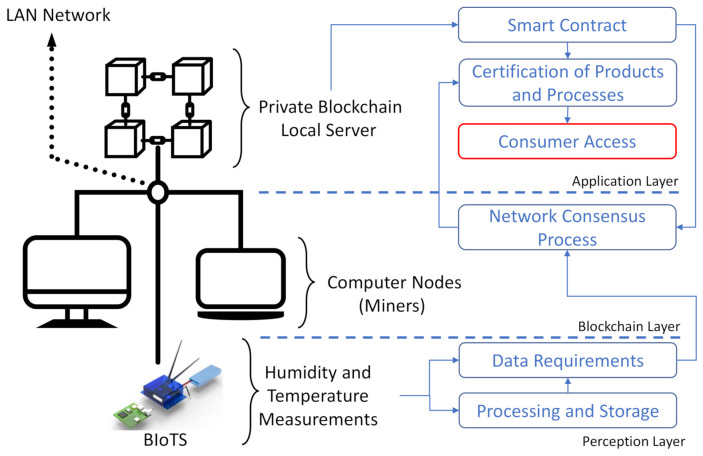
Evaluation scenario

**Figure 12 sensors-21-04388-f012:**
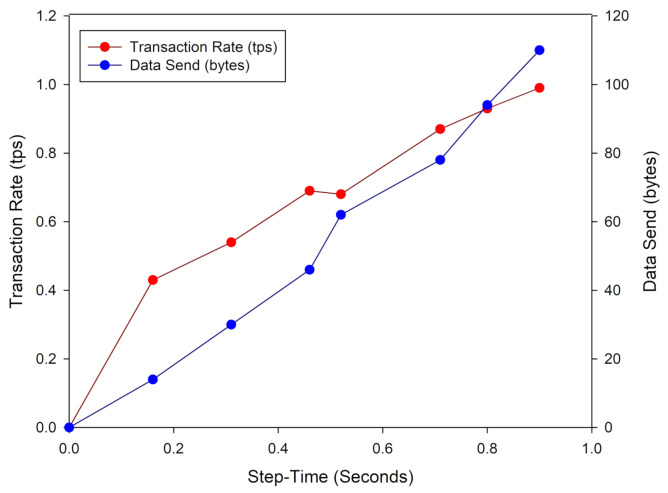
Transaction rate and data size sent by BIoTS.

**Figure 13 sensors-21-04388-f013:**
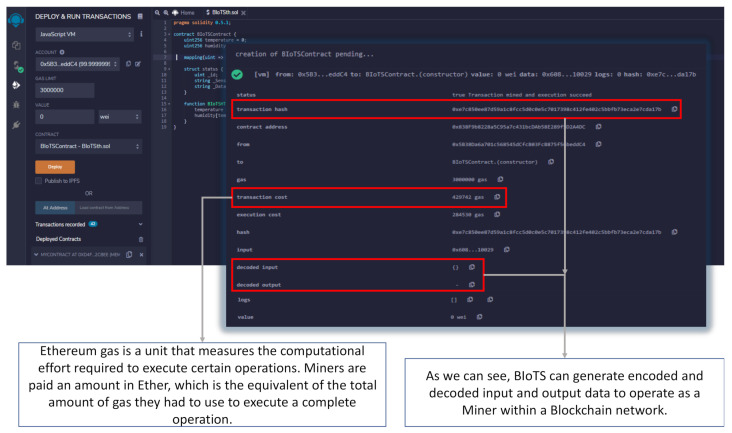
Transaction made by BIoTS in Blockchain Ethereum.

**Table 1 sensors-21-04388-t001:** Security issues, threats, and technologies.

Architecture Layer	Threats in Security	Weaknesses	Related Works	Attacks
**Application Layer**				
Internet	Confidentiality	Access Centralization	[25,26,27,28,29]	Pishing, Malware
**Transport Layer**				
Wireless	Rogue access points, Misconfiguration	Hacking, Signal lost	[30,31,32,33,33]	DoS, War dialing, protocol tunneling; man-in-the-middle
**Perception Layer**				
Sensor Nodes	DoS, Exhaustion, Unfairness, Sybil	Flooding, Routing Protocols	[30,34,35,36,37,38,39]	Jamming, Tampering, Collisions

**Table 2 sensors-21-04388-t002:** Generic Features Analysis of Consensus Algorithms (based on [17,57]).

Consensus Algorithm	Blockchain Type	Mining	Consensus Category	Reference	Experiment Setup	Communication Model	Energy Consumption
PoW	Permission-less	Based on computational power	Proof-based	[59]	Real implementation	Asynchronous	538 KWh
Implicit Consensus	Permissioned	Proof based mining	Proof-based	[60]	Theoretically evaluated	Asynchronous	Unknow
PoV	Consortium	Vote-based mining	Vote-based	[61]	Simulation, Single machine	-	Unknow
Ripple	Permissioned	Vote-based mining	Vote-based	[62]	Simulation, Single machine	Asynchronous	Unknow
DBFT	Permissioned	Non-proof of work based mining	Vote-based	[63]	Proposed solution is not validated through experiments	Asynchronous	Unknow
PoT	Permission-based consortium	Probability and vote based mining	Vote-based	[64]	Simulation, Single machine	Asynchronous	Unknow

**Table 3 sensors-21-04388-t003:** Logic elements used on DE0-Nano FPGA.

FPGA	Total Logic Elements	Percentage Available
DE0-Nano	22,320	100%
**Block**	**Total Logic Elements**	**Percentage Used**
SHA-256 and PoW	10,347	46%
I2C-Master	168	≤1%
I2C-Slave	114	≤1%
SD-CARD	289	1%
Total Area Used	10,556	47%

**Table 4 sensors-21-04388-t004:** Evaluated parameters.

Step-Time	Transaction Rate	Data Send	Latency
**(seconds)**	**(tps)**	**(bytes)**	**(seconds)**
0.00	0	0	0
0.16	0.43	14	0.03
0.31	0.54	30	0.05
0.46	0.69	46	0.06
0.52	0.68	62	0.07
0.71	0.87	78	0.09
0.8	0.93	94	0.13
0.9	0.99	110	0.103

**Table 5 sensors-21-04388-t005:** BIoTS-Security behavior.

Attack	Description	Attack likelihood	Resistance to Attack
Sensor Tampering	Manipulate sensors to acquire data readings	Unlikely	High
Sensor Feed Modification	Modify the sensor feed and firmware during communications process	Possible	High
Sybil Attack	Creates multiple identities and manipulates the device’s reputation.	Unlikely	High
DoS, Protocol tunneling; man-in-the-middle	Shut down a machine or network and The attacker sets up rogue hardware pretending to be a trusted network as Wi-Fi	Unlikely	High
Jamming, Collisions	Is an attempt to find two input strings of a hash function that produce the same hash result	Possible	Moderate

## Data Availability

Not applicable.

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
