# Peer review of "Blockchain-IoT Sensor (BIoTS): A Solution to IoT-Ecosystems Security Issues"

_sensors, 2021, doi:10.3390/s21134388_

Round 1
Reviewer 1 Report
-The Result and Discussion part needs to be modified. In particular, the evaluation should focus on the Food Traceability scenario for the experimental environment. -Need to specify the performance verification results for BIoTS. -The number system, text error, etc. needs to be corrected. -For example, too many descriptions of existing results such as SHA-256 algorithm, PoW consensus algorithm, etc. are not suitable for paper construction. -Additional VHDL module descriptions are required, and it is expected that the processing speed will be slow to operate consensus algorithms such as PoW on resource-constrained sensors. Further consideration is needed for alternatives to this.
Reviewer 2 Report
The paper describe an architechture and implementation of a Blockchain-enabled IoT device sending temperature and humitity as an usecase. The paper describes an interesting approach which is worth for publication. But the paper has to be improved in several aspects, which is listed as follows:
General:
- related work is missing.
- Generally a summeray of the evaluation part in the conclusion of the paper is missing
- the approach uses PoW why? PoS is offered by Ethereum and woud be more enery efficient.
- 2.2.3. BIoTS Consensus Algorithm (PoW): The most interesting part for this paper is quit short. It is unclear, how you solved it. Please add a lot more detials. E.g. did you implement it direct on the device? How about the energy consumption?
- structure of the paper hast to be improved: e.g. introduction, related work, use case, BIoTs architecture, implementation details, evaluation, conclustion
- you should have a look for the capitalization of words. Very often you should not use capital letters.
- Fig. 6 should be moved at the beginning. It clearifies a lot. It gives an overview of the strucutre and it can be always used to explain the contribution.
Abstract:
The abstract is an introduction, but the abstract about the content of the paper is missing.
Introduction:
- p.2,l.37: "Social, because the information transparency concept
is assured through validated blocks by participants on the network"
I would not call this a social aspect, when nodes are communicating using a consensus mechanism. Social means relating to society or to the way society is organized. I do not see yet how sensors are a social organisation. Make it more clear or remove it.
Materials and Methods
- the title is not appropriate. It does not tell the reader about the content of the section
- p.3,l.94: "...IoT network will not have security properties beyond those allowed by its capabilities." -> This is not a good formulation. I do not like the word. "allowed" I guess you mean "possible".
- Fig.1: You talk about security in 3 dimensions: application, user, architecture. To make the Fig. consise the Fig. should be made 3 dimensional.
Then you have a dimension applicaiton, archtiecture and user.
- The categorization of: access, transparency, and integrity does not fit.
For example, in the row transparency there are security issues like: access attack, DoS, ... This does not fit.
I suggest to redraw the figure and reduce the information and do rather several figures. It seems the author wanted to put everything in one figure, but the result contains errors, which are not resolvable in chosen layout.
- p.3,l.112: the list is an intro sentence missing. The labels: "Point of view 1" hast to be improved. The term is too general. Maybe: "IoT Architecture View" for example.
- p.4,l.123: The senctence: "The issues identified here are not part of the security problems faced by Blockchain" This is not understandable. What issues do you mean?
- p.4,l.126: Now a list of selected protocols are stated. But why exactly these protocols. Bevor you list a lot more.
- p.4,l.131: Again there is an introduction missing. What is the purpose of that numbered list?
- p.4,l.131: Sentence: "differentiation of problems according to the nature of the network" This is a wrong assumption. You can have the security issues, like MITM at all layers. Maybe you can argue, that some are more propable then others. But you have to justify this.
- p.5, table 1: Suddenly in the application layer Blockchain is appearing. I do not think that Blockchain is an application is part of an application. You have to explain what you mean.
- 2.1.2. Security Solutions on The IoT Systems -> after I read the section, I think the title is not appropiate. You propably want to say: "Blockchain-based approaches to Taggle IoT Security
- p.6,l.214: I think a item like: vii) securit is missing. Since you explicitly want to integrate blockchain, I think this feature is essential.
- p.6,l.218: "Below block B" -> above
- minor issue: Why mention first block B and then A?
- Fig.2 A+B): the blocks have to be described in more detail.
Most interesting is the integration of Proof of Work into the HW of the device.
It is essential, that you describe the integration with blockchain.
- Fig.2 A) you give an archtiectur overview of the Ethereum Blockchain. Can you reference it.
- p7,l.264: "time-stamp" how do you synchronize the time-stamp?
- p8,l283: "448mod512" -> spaces missing
- p9, Algorithm 1 SHA-256: Reference is missing
- 2.2.3. BIoTS Consensus Algorithm (PoW): The most interesting part for this paper is quit short. It is unclear, how you solved it. Please add a lot more detials. E.g. did you implement it direct on the device? How about the energy consumption?
Results and Discussion
- Fig.9: Could you add a picture of the real hardware?
- table 4 is interesting, but the content is not explained and therefore not clear how the author came to the results.
Conclusions:
- To reference to figures in a conclusion section is not adviasable. Please give an overall statement about the goals reached in this work.
- p.15,l447: "good performance", what does this mean -> Please, quantiative numbers
- Generally a summeray of the evaluation part of the paper is missing
Round 2
Reviewer 1 Report
- It needs to be re-organized to focus on security issues, threats from a BIoTS perspective.
- A detailed analysis of the evaluation of computational performance problems in implementing the PoW consensus algorithm of BIoTS is required. In particular, we look forward to a comparative review of consensus algorithms suitable for BIoTS, such as PoS, DPoS, and PBFT.
- Please clean up the connectivity between Table 1 and Figure 2.
- A review of the evaluation scenarios, particularly those of the BIoTS Security Behavior, is required.
- Please re-configure Figure 12 with Figure 3.
- Please reconfirm the spell error in the body.
-Figure 5 : Security Requeriments ---> Requirements
-BIoTS Cryptographyc Algorithm ---> Cryptographic
Reviewer 2 Report
A lot has been improved. Just some minor issues:
Use Case:
* better title: -> Foodchain Tracebility Using Blockchain
5. Implementation Details:
Numbering is wrong. 5.0.1 -> 5.1 and 5.0.2 -> 5.2 ...
Round 3
Reviewer 1 Report
Please reconfirm the composition, experiment, etc. of the manuscript.